# ^11^C-Methionine PET/CT in Assessment of Multiple Myeloma Patients: Comparison to ^18^F-FDG PET/CT and Prognostic Value

**DOI:** 10.3390/ijms23179895

**Published:** 2022-08-31

**Authors:** Maria I. Morales-Lozano, Paula Rodriguez-Otero, Lidia Sancho, Jorge M. Nuñez-Cordoba, Elena Prieto, Maria Marcos-Jubilar, Juan J. Rosales, Ana Alfonso, Edgar F. Guillen, Jesus San-Miguel, Maria J. Garcia-Velloso

**Affiliations:** 1Department of Nuclear Medicine, Clinica Universidad de Navarra, CCUN Applied Medical Research, Instituto de Investigación Sanitaria de Navarra, 31008 Pamplona, Spain; 2Department of Haematology, Clínica Universidad de Navarra, CCUN Applied Medical Research, Instituto de Investigación Sanitaria de Navarra, 31008 Pamplona, Spain; 3Research Support Service, Central Clinical Trials Unit, Clinica Universidad de Navarra, 31008 Pamplona, Spain

**Keywords:** multiple myeloma, positron emission tomography, PET/CT, ^11^C-methionine, ^18^F-FDG, prognosis, volume-based parameters, TMTV, TLG, TLMU

## Abstract

Multiple myeloma (MM) is the second most common haematological malignancy and remains incurable despite therapeutic advances. ^18^F-FDG (FDG) PET/CT is a relevant tool MM for staging and it is the reference imaging technique for treatment evaluation. However, it has limitations, and investigation of other PET tracers is required. Preliminary results with L-methyl-[^11^C]- methionine (MET), suggest higher sensitivity than ^18^F-FDG. This study aimed to compare the diagnostic accuracy and prognostic value of ^1^FDG and MET in MM patients. We prospectively compared FDG and MET PET/CT for assessment of bone disease and extramedullary disease (EMD) in a series of 52 consecutive patients (8 smoldering MM, 18 newly diagnosed MM and 26 relapsed MM patients). Bone marrow (BM) uptake patterns and the detection of focal lesions (FLs) and EMD were compared. Furthermore, FDG PET parameters with known MM prognostic value were explored for both tracers, as well as total lesion MET uptake (TLMU). Median patient age was 61 years (range, 37–83 years), 54% were male, 13% of them were in stage ISS (International Staging System) III, and 31% had high-risk cytogenetics. FDG PET/CT did not detect active disease in 6 patients, while they were shown to be positive by MET PET/CT. Additionally, MET PET/CT identified a higher number of FLs than FDG in more than half of the patients (63%). For prognostication we focussed on the relapsed cohort, due to the low number of progressions in the two other cohorts. Upon using FDG PET/CT in relapsed patients, the presence of more than 3 FLs (HR 4.61, *p* = 0.056), more than 10 FLs (HR 5.65, *p* = 0.013), total metabolic tumor volume (TMTV) p50 (HR 4.91, *p* = 0.049) or TMTV p75 (HR 5.32, *p* = 0.016) were associated with adverse prognosis. In MET PET/CT analysis, TMTV p50 (HR 4.71, *p* = 0.056), TMTV p75 (HR 6.27, *p* = 0.007), TLMU p50 (HR 8.8, *p* = 0.04) and TLMU p75 (HR 6.3, *p* = 0.007) adversely affected PFS. This study confirmed the diagnostic and prognostic value of FDG in MM. In addition, it highlights that MET has higher sensitivity than FDG PET/CT for detection of myeloma lesions, including FLs. Moreover, we show, for the first time, the prognostic value of TMTV and TLMU MET PET/CT in the imaging evaluation of MM patients.

## 1. Introduction

Globally, multiple myeloma (MM) is the most frequent Plasma cell malignancy (PCM), the second most frequent haematological malignancy and represents 1% of all cancers [1,2]. Despite important therapeutic advances, that significantly improve life expectancy, most patients relapse, with an estimated mean death rate of 3.4 per 100,000 people per year [3].

The diagnosis of overt myeloma is based on the presence of at least 10% of clonal bone marrow plasma cell (BMPC) and serum M-protein or urinary monoclonal component, together with the presence of myeloma-related organ dysfunction, including increased calcium level, renal dysfunction, anaemia, and bone lesions (CRAB criteria), or the recently defined as SLIM CRAB criteria, that include clonal BMPC infiltration >60%, involved/uninvolved free light chain (FLC) ratio >100 (involved FLC level must be >100 mg/L) and >1 focal lesion (FL) on Magnetic Resonance Imaging (MRI) larger than 5 mm [4,5]. Recommended diagnostic tests [6] include routine laboratory tests, BM examination and imaging techniques, such as whole-body X-ray (WBXR), MRI, computed tomography (CT) and molecular imaging using Positron emission tomography (PET). In 2014, the International Myeloma Working Group (IMWG) established the diagnosis of bone lesions [5] according to the presence of >1 lytic lesion in CT or PET/CT, regardless of whether they were detected on WBXR, or >1 unequivocal FL >5 mm in MRI.

PET/CT with radiolabelled glucose analog [^18^F]-2′-deoxy-2′-fluoro-D-glucose (FDG), is a reliable non-invasive imaging technique with global sensitivity of 90% and specificity ranging from 70 to 100% for staging MM [7], and has demonstrated prognostic impact not only on staging, but also on response assessment [8,9,10].

The studies of Bartel et al. [8] found that the presence of ≥4 FL was associated to shorter overall survival (OS) and progression free survival (PFS). Subsequently, Zamagni et al. [9] demonstrated poor prognosis associated with ≥4 FL, maximum Standardized Uptake Values (SUV_max_) >4.2 and the presence of EMD, the latter with prognostic value on PFS and OS, as recently confirmed by Moreau et al. [10].

Given that metabolism is considered a hallmark of cancer and glycolysis is one of the main metabolic pathways that are deregulated [11], new metabolic biomarkers, such as total lesion glycolysis (TLG) and total metabolically active tumor lesion volume (TMTV) have been investigated [12,13]. However, the diagnostic value of FDG decreases when only diffuse BM infiltration or only small lytic skeletal lesions (smaller than 10 mm) are present [14,15,16].

Methionine, labeled with C-11 (MET), is an amino acid PET tracer used mainly in oncological diseases of the central nervous system. The rationale for its use in MM is related to the evidence that radiolabeled amino acids show rapid metabolic absorption and incorporation into newly synthesized immunoglobulins. MET appears to be the best radiopharmaceutical currently, apart from FDG, for evaluating MM, with higher sensitivity in detecting focal lesions and extramedullary disease and better correlation with tumor burden [17,18]. Preliminary data suggest that it may be superior to ^18^F-FDG-based imaging, even if the literature is very limited, and, therefore, insufficient for the use of this tracer in clinical practice. [19,20,21,22]. On the other hand, despite the fact that recent publications show a good correlation with other well-known myeloma prognostic factors [23], the prognostic value of MET parameters has not yet been clarified.

The purposes of the present study were to compare MET and FDG PET/CT in the evaluation of active disease at staging and re-staging MM, to analyze the correlation between the BM uptake patterns and BM infiltration, and to evaluate the prognostic impact of FDG and MET PET/CT in patients with myeloma.

## 2. Results

### 2.1. Patients’ Characteristics

Fifty-two patients with overt MM (n = 44) or SMM (n = 8) were included. From the 44 MM cases, 18 corresponded to newly diagnosed patients (NDMM), while the remaining 26 were referred for re-staging, due to progression/biological relapse. In 46 out of 52 patients (88.5%), information about BMPC infiltration was available. Median (IQR) values were 20.50 (8–40). Sixteen patients (31%) presented high-risk cytogenetics, defined as del17p (tp53), and/or t(4;14) and/or t(14;16). Patients’ characteristics are summarized in Table 1.

The majority of relapsed patients had been treated with proteasome inhibitors (24/26, 92.3%) and/or immunomodulatory drugs (20/26, 76.9%), as well as monoclonal antibodies (9/26, 34.6%) and ASCT (5/26, 19.2%).

### 2.2. MET and FDG PET-CT Findings

#### 2.2.1. Patient-Based Analysis

At baseline, 46 patients were concordant for both tracers, 44 as positive (4 SMM and 40 MM) and 2 SMM as negative, while 6 cases were discordant, due to a positivity of MET without correlation on FDG (*p* < 0.001) (Figure 1). The Kappa index between both tracers was weak (κ: 0.361, *p* < 0.01).

Interestingly, 28 out of 44 patients with positivity of both tracers had a higher number of FLs in MET PET/CT, while the remaining 16 patients had the same number of FLs (Figure 2). The 6 cases with MET/FDG discrepancies corresponded to 3 patients with NDMM, 1 MM relapsed and 2 patients with high-risk SMM. (Figure 3 and Figure 4).

#### 2.2.2. PET Bone Patterns-Based Analysis

FDG PET/CT was positive in 44 patients, 2/44 patients (5%) showed diffuse BM infiltration, 17/44 patients showed only FLs (39%), the remaining 24/44 patients (55%) showed a mixed pattern, and 1/44 patient presented only with EMD. In MET PET/CT, 5/50 patients (10%) showed diffuse BM infiltration as the only finding, 4/50 patients (8%) showed FLs and 41/50 (82%) showed a mixed pattern with diffuse BM infiltration and FLs. Therefore, MET PET/CT detected BM diffuse infiltration in a greater number of patients, not only as a purely diffuse pattern (10% vs. 5%) but also as a mixed pattern with FLs and BM infiltration (82% vs. 55%). The Kappa index between tracers was moderate in the case of FLs (κ: 0.468, *p* < 0.01), and poor/fair in the case of diffuse and mixed patterns (κ: 0.192 and 0.227, *p* = 0.009 and 0.018, respectively). Table 2, Figure 5.

#### 2.2.3. Lesion-Based Analysis

On a lesion-based analysis, imaging with MET identified more FLs than FDG in 33 patients (63%, *p* = 0.005). In fact, it detected 1–3 FLs in 10 patients, 4–10 FLs in 11 patients and >10 FLs in 24 patients. On the other hand, FDG PET/CT identified 1–3 FLs in 14 patients, 4–10 FLs in 9 patients and > 10 FLs in 18 patients. The kappa index between the two tracers for the detection of FLs was moderate (κ = 0.637, *p* < 0.001), according to the Deauville criteria (1 patient at staging, 2 at restaging), while most of the FLs had a score of 4 (42.3%) (15 patients at staging, 7 patients at restaging) or 5 (30.8%) (8 patients at staging and the same number at restaging). EMD was detected in 7 patients with both tracers, but MET PET/CT detected more lesions in 2 patients in 7 locations (lung, pleura, lymph nodes, kidneys, muscle, soft tissues and periorbital fat) (Figure 6). Median (IQR) values for BM SUV_max_ and FL SUV_max_ were 2.92 (2.01–3.63) and 6.11 (3.34–8.73), respectively, on FDG PET/CT, and 4.81 (3.98–7.16) and 7.83 (5.5–10.64) on MET PET/CT, being significantly higher (*p* < 0.001).

According to the diffuse infiltration of BM in FDG PET/CT study, 2 patients were categorized as Deauville 1 (no uptake), 7 patients as Deauville 2 (uptake below/equal the mediastinum blood pool), 17 patients as Deauville 3 (uptake higher than mediastinum but below/equal the liver uptake), 20 patients as Deauville 4 (uptake higher than the liver’s) and 6 patients as Deauville 5 (2.5 times the liver uptake). Deauville > 4 was considered to be positive (Table 3).

#### 2.2.4. Correlation of Tracer Uptake with Bone Marrow Involvement

Furthermore, the correlation between BMPC infiltration rates and BM uptake on FDG and MET PET/CT was explored. First, the median SUV_max_ value of lumbar BM uptake was measured with both techniques, being 4.70 (IQR 1.32–12.04) on MET PET/CT, superior to FDG PET/CT (2.66, IQR 0.93–6.91, *p* < 0.001). In the iliac crest, median SUV_max_ was also statistically superior in MET PET/CT (3.98, IQR 1.45–9.64) than in FDG PET/CT (2.32, IQR 0.87–5.64, *p* < 0.001). Overall, moderate correlation between MET and FDG was found for lumbar and iliac SUV_max_ tracers [r = 0.73 (*p* < 0.001) and r = 0.59 (*p* < 0.001), respectively.

Upon analysing the correlation of semiquantitative parameters by FDG PET/CT in lumbar/iliac/BM region with bone marrow infiltration, no correlation was found. Whereas for MET PET/CT a low correlation was observed between the BM SUV_peak_ and BM infiltration (r = 0.30, *p* = 0.04).

Tumor to background ratio (TBR) values (SUV_max_/Liver SUV_max_ ratio) for MET were 0.50 + 0.27 for the vertebral BM and 0.42 + 0.22 for the iliac. It is important to highlight that the TBR was significantly correlated with the percentage of BM infiltration, moderately in the spine (r = 0.56, *p* < 0.001) and in the iliac bone (r = 0.48, *p* < 0.001), the last one of higher interest, given that it is the most common site of BM biopsy for the evaluation of BMPC infiltration.

#### 2.2.5. Prognostic Value of MET and FDG PET/CT

For this objective, we excluded the SMM patients because none of them had progressed, and, thereby, focusing the analysis on the remaining 44 patients with active MM.

In FDG PET/CT, 39/44 patients (89.6%) had focal lesions: 1–3 FLs were observed in 12/44 patients (27.3%) and ≥4 FLs in 27/44 patients (61.4%); the remaining 5 patients did not show any FLs. SUV_max_ > 3.9 was observed in 30/44 patients (68%) and TMTV > 210 cm^3^ was present in 13/44 patients (29.5%), while TLG > 205 g or >620 g were observed in 50% (22/44) and 31.8% (14/44), respectively. Median TMTV and TLG were 86.1 cm^3^ (IQR 38.9–320.1) and 267.2 (IQR 98.7–1099.9), respectively.

In MET PET/CT, all but one patient (43/44; 97.7%) had detectable FLs, 1–3 FLs were present in 9/44 patients (20.5%) and ≥4 FLs in 34/44 (77.3%). Median TMTV was 329.3 cm^3^ (IQR 164.4–651.7), while for TLMU the median was 1355.37 (IQR 758–3241) (Figure 7).

During follow-up, 15 patients had progressed and 8 patients had died. The median follow-up of this series was 15.7 months (IQR 7.8–26.9). Nevertheless, according to the different expected behaviour of relapsed and NDMM patients, the majority of the 15 progressions had occurred in the relapsing patient cohort (11 vs. 4 progressions in the NDMM cohort).

When the PET-CT biomarkers were analyzed in the NDMM group (4 progressions out of the 18 patients), 75% of patients who progressed had ≥4 FLs on FDG and MET PET/CT, as is shown in Table 4. TLG FDG and TMTV FDG median values were higher, and TLMU MET and TMTV MET were significantly higher than in patients who did not progress. Regarding relapsed patients (14 out of 26 have already progressed) a similar picture was observed (Table 5), 72.7% of patients who progressed had ≥ 4 FLs on FDG and 81.8% on MET PET/CT; the median values for TMTV FDG, TMTV MET, TLG FDG, and TLMU MET were significantly higher than in patients who did not progress (Table 5). Of note, the presence of EMD did not reach statistical significance with any of the two tracers, which could be due to the low number of cases with EMD in this series.

For the univariate analysis we decided to focus on the relapsed patients, due to the low number of events observed in NDMM patients. A significant association was found between PFS and the following FDG PET/CT parameters: number of FLs > 10, TMTV, and TLG, while FLs ≥ 4 did not reach statistical significance (Table 6). Regarding MET PET/CT, a significant association was found between PFS and the following MET PET/CT parameters: TMTV, and TLMU; nonetheless, the presence of >10 FLs by MET did not influence outcome, probably because a higher threshold was needed for prognostication upon using this tracer.

## 3. Discussion

FDG PET/CT represents one of the most valuable tools to evaluate disease burden in patients with MM and has demonstrated prognostic impact, not only in staging but also in response assessment [8,9,10]. However, there is evidence that more sensitive and specific tracers are required. This prospective study was conducted to evaluate the outcome of FDG and MET PET/CT in patients with MM. To the best of our knowledge, our study is the first to evaluate the prognostic value of tumor burden biomarkers by MET PET/CT in patients with MM.

In the present study, we found that MET had higher detectability than ^18^F-FDG, with negative FDG PET/CT uptake and detectable disease in MET PET/CT in six patients (27%). These results are comparable to previous publications [19,20,21,22] that reveal a not negligible 10–15% of MM patients without FDG avidity [24]. Two out of six patients with active disease on MET PET/CT without correlation with FDG PET/CT corresponded to patients with high-risk MM. In fact, these patients with a normal pattern on FDG PET/CT and active disease on MET PET/CT showed worse evolution than those patients that presented negative results on the dual tracer study, progressing at 9 and 42 months after staging, respectively.

In parallel to previous publications that demonstrate a higher sensitivity of MET PET/CT, in the present study, this tracer suggested higher detectability than FDG PET/CT, with a greater number of focal lesions in 63% of patients and more extramedullary lesions detected in 25% of patients, located in lungs, pleura, kidneys, etc. Unfortunately, the biological implications of these discrepancies remain unexplained by the absence of biopsies to analyze their molecular nature. In the literature, only one publication [22] confirms the higher specificity of MET by biopsy, although the authors recognised the small sample size and the lack of study of the underlying biological factors.

The fact that the infiltration rate of BMPC is positively correlated only with PET biomarkers upon suing MET PET/CT, could indicate that this tracer better reflects total tumor burden, as recently postulated by Lapa et al. [21]. These authors found that MET was positive in patients with non-secretor MM, and SUV_mean_ and SUV_peak_ demonstrated superior grade of correlation with the BM infiltration than FDG, suggesting that the aminoacidic tracer uptake could represent the tumor anabolism and then, better reflect the true total tumor burden.

Regarding the prognostic value of PET/CT, the impact of the number of focal lesions in FDG PET/CT has also been confirmed in this study. Paschali et al. prospectively analyzed FDG PET/CT in patients with newly diagnosed and relapsed/refractory MM [25]. The presence of ≥10 focal lesions negatively predicted for overall response. Moreover, we showed the prognostic impact of TLG and TMTV FDG, an observation that was coincidental with that reported by McDonald et al. [13].

For the first time, MET PET/CT biomarkers and their prognostic implications have been assessed, being the first study to date that evaluates the prognostic implications of the volume-based parameters of dual tracers PET/CT in multiple myelomas. Our data indicate that TMTV MET and TLMU may have prognostic implications. In the case of MET PET/CT, the lack of prognostic relevance of the number of focal lesions and SUV_max_ values could suggest that higher cut-off points should be explored for MET, due to its higher sensitivity. This seems a logical argument, given that the semi-quantitative parameters of MET are usually superior to FDG PET/CT, not only in physiological circumstances, but also when there is tumor infiltration, either focal or diffuse. For this reason, the search for specific thresholds for MET PET/CT is mandatory to confirm whether this tracer has prognostic value.

Regarding the new tumor biomarkers, our study confirms the prognostic value of TMTV and TLG FDG PET/CT, as described by Mc Donald et al. [13]. As regards MET PET/CT biomarkers with prognostic value, we cannot compare our results with other groups as it is the first time that the prognostic impact of this tracer has been proposed.

The new tumor biomarkers have advantages over the manual determination of SUV, since it is a more automatic process that provides not only the total tumor burden in PET/CT, but also the most important semi-quantitative parameters of the hottest focal lesion as SUV_max_, SUV_mean_, and SUV_peak_. However, certain considerations are necessary before its extended use. As our group has postulated in a previous work, there is still a need to standardize the calculation of TMTV [23]. However, the aim of the present study was to analyze the prognostic implications of the new tumor biomarkers, focusing especially on MET PET/CT, so it was out of the scope to describe procedural aspects, the object of study of the aforementioned publication.

## 4. Materials and Methods

### 4.1. Patients

From August 2015 to May 2019, a total of fifty-two consecutive patients, 44 with MM and 8 with Smoldering MM (SMM), referred for dual tracer FDG and MET PET/CT for staging (n = 26) and restaging (n = 26) were enrolled prospectively.

### 4.2. Methods FDG and MET- PET/CT Methodology and Evaluation

FDG and MET were synthesized in-house with an 18 MeV Cyclotron (Navarra; Cyclone 18/9, IBA Radio pharma Solutions, Louvain-la-Neuve, Belgium) [26]. PET/CT was performed on a PET/CT scanner (Siemens Biograph mCT 64, Siemens, Knoxville, TX, USA) within a median interval of 1 day between FDG and MET scans (range, 0–15). Patients fasted at least 4 h before FDG (3 to 5 MBq/kg) and MET injection (6–10 MBq/kg). No adverse effects associated to radiotracer injection were observed. PET/CT scans were acquired after 60 min (FDG) or 20 min (MET), using non-contrast-enhanced CT with 120 kV Care Dose 4D and a quality reference of 80 mAs, including the skull to the proximal thighs and lower limbs. Consecutively, PET emission data were acquired in 3D-mode with an emission time of 2–3 min per bed position in the skull to mid-thighs and 1 min per bed in the lower limbs. For each acquisition, two different reconstructions were performed: a high-quality reconstruction optimized for lesion detection and tumour volume segmentation (PSF + TOF, Gaussian filter 2 mm) and a reconstruction optimized for quantification (OSEM with 2 iterations, 8 subsets and a Gaussian filter of 8 mm for studies before July 2016 and OSEM with 3 iterations, 21 subsets and a Gaussian filter of 5mm from July 2016 onwards, in compliance with EARL).

After anonymization, both FDG and MET PET/CT were visually and semi-quantitatively analyzed by two PET professionals (junior -MML- and senior expert with more than 20 years of experience-MJGV-) on a dedicated workstation with Syngo.via (Siemens) PET software applying the criteria to define FDG PET/CT positivity described by Moreau et al. [10].

First, a visual inspection of scans was performed. BM involvement was defined as a homogenous FDG uptake in the axial and appendicular skeleton higher than the liver activity or as a heterogeneous uptake regardless of the intensity of uptake. FL was defined as the presence of areas of increased focal tracer uptake within the bones, more intense than the normal background uptake of BM, recorded according to the International Myeloma for PET Use criteria (IMPeTUs) [27], and EMD was defined as FDG-avid tissue in soft tissue not contiguous to bone on CT examination. After treatment, the normalization of FDG PET/CT uptake was defined as residual uptake greater than or equal to liver activity in FL, BM, and EMD. All other PET/CT results were considered positive, excluding increased FDG uptake related to bone reconstruction or to the use of hematopoietic growth factors.

In MET PET/CT studies, FLs were defined as bone lesions with higher uptake than the surrounding activity in normal tissue or contralateral structures, as previously described [22,23], and BM involvement was defined as increased uptake in the BM with/without expansion to the distal part of large bones. Given the absence of standardized criteria for their interpretation, the same variables obtained for FDG PET/CT were collected for further comparisons. Since BM uptake is physiologically superior to ^18^F-FDG, tumor to background ratios (TBR) were calculated in quantification reconstruction (OSEM 2.8 or OSEM 3.21 depending on the acquisition date), liver uptake being the reference organ, as in IMPeTUs criteria.

For semi-quantitative analysis, tracer uptake was determined by drawing a volume of interest (VOI) within the medullary and extramedullary lesions. In addition, a volume of interest was placed on the iliac crest and lumbar spine (L2), as long as no evident lytic lesions were observed in order to correlate the results to the BM biopsy. Other regions included as reference organs were the right hepatic lobe and the lumen of the descendant aorta, avoiding areas of calcification in the vessel wall. The semi-quantitative parameters recorded were SUV_max_ and SUV_peak_ of FL, BM and reference organs. As in the TBR calculations, the reconstructions used for quantification were OSEM 2.8 or OSEM 3.21.

In addition, the FDG and MET PET/CT images were also subjected to a 3-dimensional volume of interest analysis of the axial and appendicular skeleton using “PET/CT Viewer Beth Israel for FIJI”, a free software available for reading and image post-processing. This software automatically delineates areas with high uptake. Then, the operator discards those physiological uptakes (bladder, brain, liver, etc.) and selects the threshold to refine the delimitation of each focus. In order to calculate the tumour volume, apart from the recommended European Association of Nuclear Medicine threshold of 41% of SUV_max_ [28], other absolute thresholds (SUV > 2, SUV > 4) and relative thresholds (SUV > 30%, 41%, 60% of SUV_max_) were used to segment the images, choosing the threshold that best fitted with the visually identified active lesions among those automatically generated. Parameters explored were MTV, defined as sum of voxels of the active bone lesions based on absolute SUV threshold or relative threshold, and TLG, calculated with the following formula: TLG = ∑ (SUV_mean_ × MTV). It was decided to use the same cut-off of MTV and TLG in FDG PET/CT as in previous studies [13] when exploring the prognostic association. For MET PET/CT, the equivalent term for TLG was the total lesion methionine uptake (TLMU), defined as MTV multiplied by mean standardized uptake values (SUV_mean_) within the boundary [29]. The median cut-off values of MTV and TLMU were chosen on MET PET/CT to explore their prognostic relevance since there were no stablished cut-offs in the literature.

### 4.3. Statistical Analysis

Quantitative values were summarized as mean and standard deviation or median and range if appropriate. Concordance analysis were performed per patient and number of lesions, estimating the kappa concordance coefficient (κ) which was classified as poor agreement (≤0.20), fair agreement (0.21–0.40), moderate agreement (0.41–0.60), substantial agreement (0.61–0.80) and almost perfect agreement (0.81–1.00). Fisher’s exact test or chi square were used for comparisons of frequency data in independent subgroups. Pearson or Spearman correlation coefficients were used for bivariate correlations. Prognostic analyses of PFS were performed using log-rank test and Cox regression analysis. The median (p50) and the 75th percentile (p75) were used as cut-off values. All statistical tests were two-tailed and a *p* value < 0.05 was considered statistically significant. Statistical analyses were performed using SPSS version 22.0 (Inc. Chicago, IL, USA) and Stata version 14.0 (Stata Corp. 2015. Stata Statistical Software: Release 14. College Station, TX, USA: StataCorp LP).

## 5. Conclusions

Although this study has the important limitation of the number and heterogeneity of the sample of patients, so the conclusions should be interpreted as preliminary and require confirmation in a larger series of patients, the results postulate for the first time the prognostic value of MET PET/CT, while confirming the prognostic value of the new tumor biomarkers in FDG PET imaging.

MET detected tumor infiltration in 11% of patients with MM and negative FDG PET/CT, detecting a greater number of lesions in the majority of patients.

As MET showed correlation with bone marrow infiltration, it could suggest this tracer better reflects the tumor burden than ^18^F-FDG, despite the assessment, in cases with slight infiltration, possibly being limited due to the physiological medullary uptake.

According to the prognostic relevance of the FDG PET/CT biomarkers in relapsed patients, in the present work the prognostic value of TLG, TLMU, TMTV on FDG and MET PET, and >10 FL on FDG PET were confirmed.

## Figures and Tables

**Figure 1 ijms-23-09895-f001:**
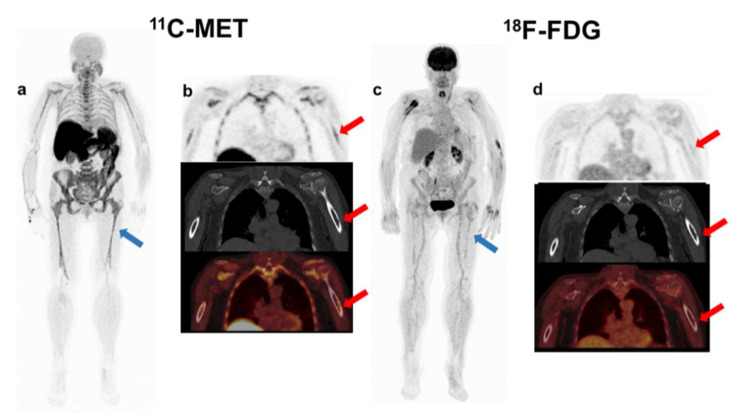
Patient with oligo-secretor Kappa MM, ISS 2. MET PET/CT showed diffuse BM uptake (blue arrow-(**a**)) and a FL with slight accumulation of the tracer in the cortical of the left humerus corresponding with a sub-centimetric lytic lesion (red arrow-(**b**)). Otherwise, FDG PET/CT demonstrated no focal accumulation in the FL (red arrow-(**d**)) and there was faint homogeneous diffuse uptake in the axial and appendicular skeleton (blue arrow-(**c**)), the intensity of which was lower than liver uptake (Deauville 3). The patient did not progress.

**Figure 2 ijms-23-09895-f002:**
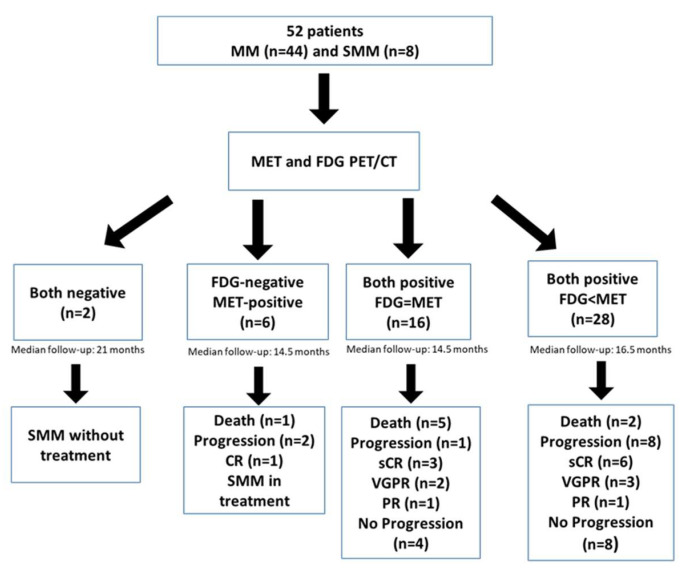
Flow chart of the study population.

**Figure 3 ijms-23-09895-f003:**
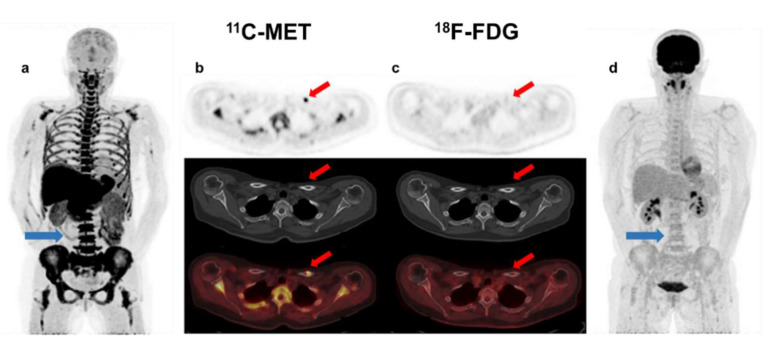
Patient with SMM with diffuse BM uptake (blue arrow-(**a**)) and FL without lytic component in the left clavicle (red arrow-(**b**)) in MET PET/CT (SUV_max_ = 1.5). This finding could represent, despite being single, a focal active lesion of multiple myeloma. However, in FDG PET/CT, BM uptake was homogeneous and lower than in the liver (blue arrow-(**d**)), so inconclusive for malignant infiltration (anaemia?), while no focal uptake was observed in the clavicle (red arrow-(**c**)).

**Figure 4 ijms-23-09895-f004:**
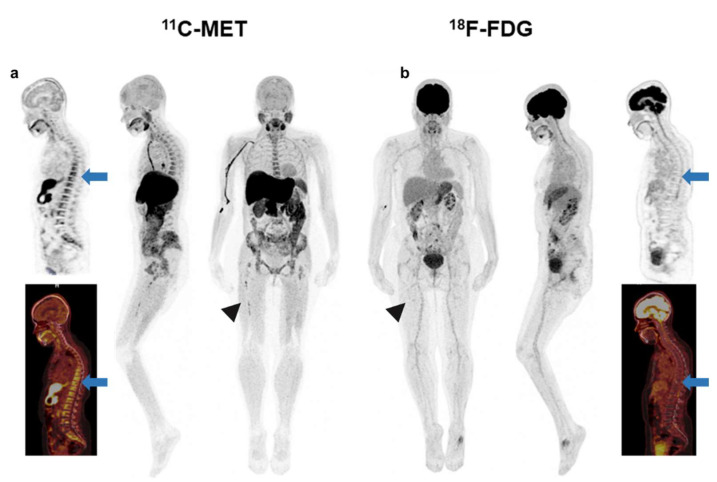
Patient with SMM with heterogeneous BM uptake in the axial (blue arrow-(**a**)) and appendicular skeleton (arrow head-(**a**)) in MET PET/CT. FDG PET/CT was negative for pathological focal tracer accumulation related to high metabolic lesions. BM uptake in the axial (blue arrow-(**b**)) and appendicular skeleton (arrow head-(**b**)) was homogeneous and lower than the mediastinal blood pool (Deauville 2). With these findings, this patient would be classified as negative or low-risk based on the FDG PET/CT and so not a candidate for the start of treatment.

**Figure 5 ijms-23-09895-f005:**
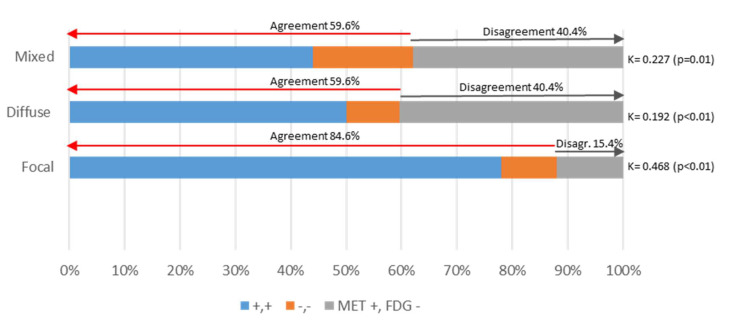
Concordance in FL and BM detection between MET and FDG PET/CT. The bar chart represents differences in the metabolic patterns detected by MET and FDG PET/CT.

**Figure 6 ijms-23-09895-f006:**
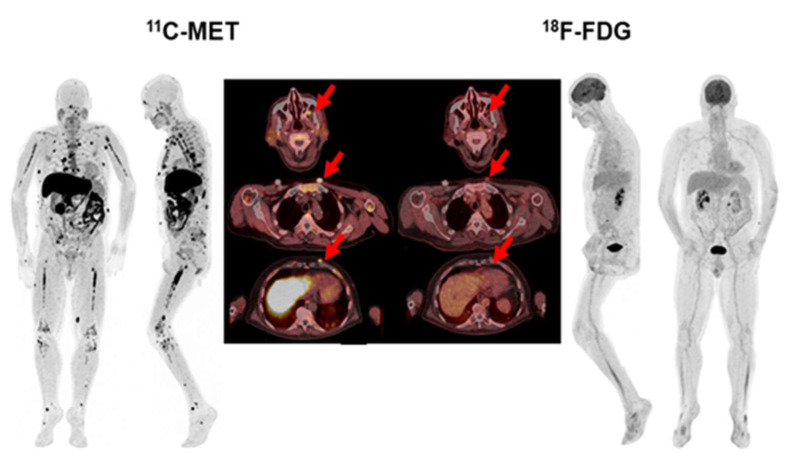
68-year-old patient with Bence Jones Lambda MM ISS-2. While FDG PET/CT showed focal uptake in subcutaneous lesions located in the chest and pulmonary nodules in lower lungs (red arrows), MET PET/CT also revealed a greater number of lung lesions, as well as pleural, lymph nodes, frontal sinus infiltration, and extensive BM infiltration.

**Figure 7 ijms-23-09895-f007:**
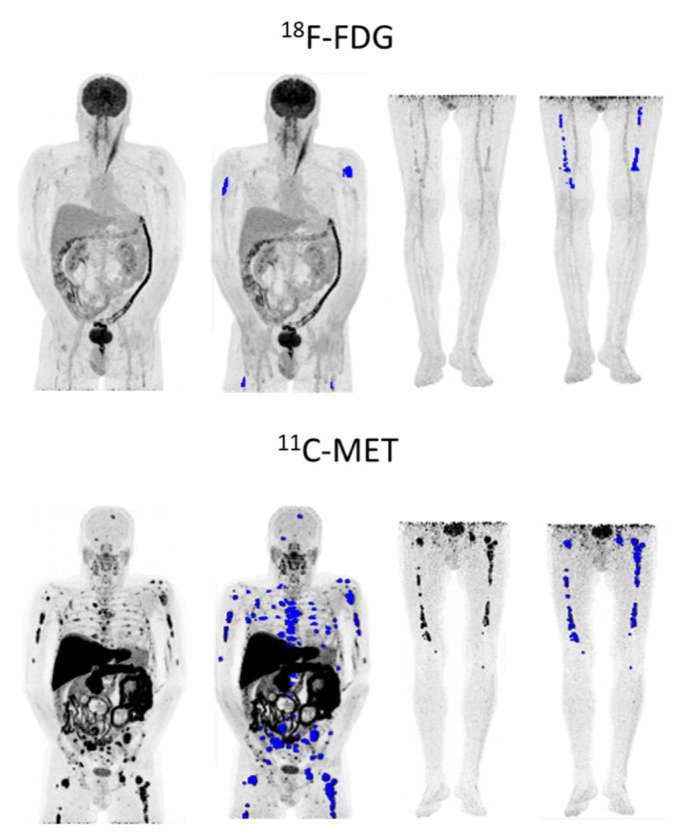
56-year-old relapsed patient with IgD lambda ISS-1 with higher tumor burden (in blue) in MET PET/CT (TMTV: 244.67 cm^3^, TLMU: 1321.86 g) than in FDG PET/CT (TMTV: 43.96 cm^3^, TLG: 76.77 g). Patient progressed even though the tumor had low FDG avidity.

**Table 1 ijms-23-09895-t001:** Patients’ characteristics.

Characteristics	n
Median age, years (range)	61 (37–83)
Sex (male/female), N (%)	28/24 (54%/46%)
NDMM/Relapsed, N (%)	26/26 (50%)
Subtype: IgG/IgA/IgD/Bence Jones/Oligosecretor, N (%)	29/6/1/15/1 (56%/11%/2%/29%/2%)
Light chain subtype, kappa/lambda, N (%)	35/17 (67%/33%)
ISS, N (%)	45/52 (87%)
Stage 1	21 (47%)
Stage 2	18 (40%)
Stage 3	6 (13%)
R-ISS, N (%)	36/52 (69%)
Stage 1	10 (28%)
Stage 2	23 (64%)
Stage 3	3 (8%)
BM infiltration % (median, range)	20.50 (1–80)
B2M, mg/L (median, range)	3.16 (1.87–4.04)
LDH (median, range)	197 (101–965)
Hgb, g/dL (median, range)	12.4 (7.40–20.40)
M protein (median, range)	1.23 (2.08–8.60)
Monoclonal component (urine) (median, range)	2.05 (0–500)
Affected FLC (median, range)	42.54 (0.14–10,500)
High Risk Cytogenetics (FISH), N (%)	16/36 (31%)
t(4;14), N (%)	7/49 (14.3%)
t(14;16), N (%)	4/49 (8.2%)
Del 17p, N (%)	9/50 (18%)
Proteasome inhibitors, yes/no, N (%)	24/26 (92.3%)
Immunomodulatory drugs, yes/no, N (%)	20/26 (76.9%)
Monoclonal antibodies, yes/no, N (%)	9/26 (34.6%)
ASCT, N (%)	5/26 (19.2%)

ISS, international staging system; BM, bone marrow; B2M, beta 2 micro-globulin; LDH, lactate dehydrogenase; Hgb, haemoglobin; FLC, free light chains; FISH, fluorescence in situ hybridization; t(4;14), translocation (4;14); t(14;16), translocation (14;16); Del 17p, deletion17p; ASCT, autologous stem cell transplantation.

**Table 2 ijms-23-09895-t002:** Uptake patterns of FDG and MET PET/CT.

	FDG	MET	Kappa Index	*p* Value
PET/CT study-positive	44/52 (84.6%)	50/52 (96.2%)	0.361	*p* < 0.001
Patient-based analysis				
FL	17/44 (39%)	4/50 (8%)	0.468	*p* < 0.01
Diffuse	2/44 (5%)	5/50 (10%)	0.192	*p* = 0.009
Combined	24/44 (55%)	41/50 (82%)	0.227	*p* = 0.018
Lesion-based analysis				
1–3 FL	14/41 (34%)	10/45 (22%)		
4–10 FL	9/41 (22%)	11/45 (24%)	0.637	*p* < 0.001
>10 FL	18/41 (44%)	24/45 (53%)		

**Table 3 ijms-23-09895-t003:** FDG PET/CT characteristics.

FDG PET/CT Uptake (Deauville)	N (%)
Presence of FL	
- Score 2	2/52 (3.8%)
- Score 3	1/52 (1.9%)
- Score 4	22/52 (42.3%)
- Score 5	16/52 (30.8%)
Presence of BM uptake	
- Score 1	2/52 (3.8%)
- Score 2	7/52 (13.5%)
- Score 3	17/52 (32.7%)
- Score 4	20/52 (38.5%)
- Score 5	6/52 (11.5%)
BM SUV_max_ (median, IQR)	2.92 (2.01–3.63)
FL SUV_max_ (median, IQR)	6.19 (3.26–8.98)

FL, focal lesion; BM, bone marrow; SUV, standardized uptake value; IQR, inter-quartile range.

**Table 4 ijms-23-09895-t004:** Baseline FDG and MET parameters in NDMM according to clinical outcome (progression).

Biomarquers		Not Progressed		Progressed		*p* Value
		% or Median	p25–p75	% or Median	p25–p75	
FL FDG	0	14.3%	-	25.0%	-	0.632
	1–3	35.7%	-	0%	-	
	4–10	21.4%	-	25%	-	
	>10	28.6%	-	50.0%	-	
TLG FDG		553.2	278.3–1097.6	1990.8	688.7–5682.4	0.083
TMTV FDG		199.0	91.7–353.6	604.9	304.5–1312.0	0.083
EMD FDG		0%	-	25%	-	0.222
FL MET	0	7.1%	-	0%	-	>0.999
	1–3	35.7%	-	25%	-	
	4–10	14.3%	-	25%	-	
	>10	42.9%	-	50%	-	
TLMU MET		2004.6	1054.3–2478.8	4394.9	3164.1–11,801.1	0.026
TMTV MET		476.1	235.5–560.4	722.1	639.7–791.2	0.034
EMD MET		0%	-	25%	-	0.222

FL, focal lesions; TLG, total lesion glycolysis; TMTV, total metabolic tumor volume; EMD, extramedullary disease; TLMU, total lesion methionine uptake.

**Table 5 ijms-23-09895-t005:** Baseline FDG and MET parameters in relapsed patients according to clinical outcome (progression).

Biomarquers		Not Progressed		Progressed		*p* Value
		% or Median	p25–p75	% or Median	p25–p75	
FL-FDG	0	8.3%	-	9.1%	-	0.112
	1–3	41.7%	-	18.2%	-	
	4–10	33.3%	-	9.1%	-	
	>10	16.7%	-	63.6%	-	
TLG FDG		98.5	12.9–151.7	1215.2	184.0–1965.1	0.003
TMTV FDG		38.8	3.7–65.8	180.1	62.2–468.4	0.002
EMD FDG		16.7%	-	27.3%	-	0.640
FL-MET	0	0%	-	0%	-	0.103
	1–3	8.3%	-	18.2%	-	
	4–10	50.0%	-	9.1%	-	
	>10	41.7%	-	72.7%	-	
TLMU MET		860.2	328.2–1147.6	3304.6	1321.9–4530.6	0.010
TMTV MET		199.8	74.1–289.1	570.5	186.5–785.9	0.021
EMD MET		16.7%	-	27.3%	-	0.640

FL, focal lesions; TLG, total lesion glycolysis; TMTV, total metabolic tumor volume; EMD, extramedullary disease; TLMU, total lesion methionine uptake.

**Table 6 ijms-23-09895-t006:** Univariate analysis of Baseline FDG and MET biomarkers in clinical outcomes (PFS) in relapsed patients.

Biomarker	HR	95% CI	*p* Value
FL FDG ≥ 4	4.61	0.96–22.11	0.056
FL FDG > 10	5.65	1.45–22.07	0.013
TLG FDG > p50	6.72	1.35–33.51	0.020
TLG FDG > p75	5.32	1.37–20.68	0.016
TMTV FDG > p50	4.91	1.01–23.91	0.049
TMTV FDG > p75	5.32	1.37–20.68	0.016
EMD FDG	3.64	0.89–14.93	0.073
FL MET ≥ 4	0.63	0.13–3.07	0.569
FL MET > 10	1.73	0.43–6.94	0.440
TLMU MET > p50	8.80	1.09–70.68	0.041
TLMU MET > p75	6.27	1.64–23.99	0.007
TMTV MET > p50	4.71	0.96–22.98	0.056
TMTV MET > p75	6.27	1.64–23.99	0.007
EMD MET	3.64	0.89–14.93	0.073

PFS, progression-free survival; HR, hazard ratio; CI, confidence interval; FL, focal lesion; TLG, total lesion glycolysis; TMTV, total metabolic tumor volume; SUV, standardized uptake values; EMD, extramedullary disease; TLMU, total lesion methionine uptake.

## Data Availability

The datasets generated during and/or analyzed during the current study are available from the corresponding author on reasonable request.

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
