# Peer review of "11C-Methionine PET/CT in Assessment of Multiple Myeloma Patients: Comparison to 18F-FDG PET/CT and Prognostic Value"

_ijms, 2022, doi:10.3390/ijms23179895_

Round 1

Reviewer 1 Report

The Authors compare 11C-Methionine and 18F-FDG PET/CT in Assessment of Multiple Myeloma, evaluating their Prognostic Value, too.

It is a prospective study in a population of 52 patients.

The paper is interesting.

Some points need clarification and/or changes:

·       Figures 1-3-4 are not clear: more in-depth explanation and arrows on the images are needed

·       The discussion is poor.

·       “After anonymization, both 18F-FDG and 11C-MET PET/CT were visually and semi- quantitatively analyzed by PET experts…..”: what was the agreement between the reviewers on visual assessment?

·       TMTV calculation is difficult and the reproducibility is often suboptimal in many tumors but particularly in Multiple Myeloma: what was the agreement between the reviewers in TMTV calculation? It is important if you want to evaluate the prognostic value

Author Response

Thank you very much for your comments and observations.

As requested we have addressed all your comments. Changes in the revised version of the manuscript have been made using the "Track Changes" function of our word processing program. We really hope that the revised version of the manuscript deals with all the reviewers’ and editor concerns. Looking forward to receiving your new comments and hopeful that the editor will find the manuscript suitable for publication.

All authors have read and approved the revised version of the manuscript.

Point 1: The methods are not adequately described.

Response 1: The methods have been described in more detail in the authors' previous publications [References 22 and 23], but if the reviewer deems it convenient, the methods can be more precisely detailed.

Point 2:  The results can be improved. Figures 1-3-4 are not clear: more in-depth explanation and arrows on the images are needed.

Response 2: Following the recommendation, more in-depth explanation of Figures 1-3-4 and arrows have been added.

Point 3: After anonymization, both 18F-FDG and 11C-MET PET/CT were visually and semi- quantitatively analyzed by PET professionals …..”: What was the agreement between the reviewers on visual assessment?

Response 3: We did not work on the agreement between the reviewers, because MML is a junior Nuclear Medicine Physician (this study is part of his research for a doctoral degree) and MJGV is a Nuclear Medicine Physician with more than 20 years of experience in 18F-FDG PET/CT and 5 years of experience in 11C-MET PET/CT. During the residency period, MIML gained experience in order to be able to recognise and discard the physiological uptake and to identify BM infiltration, FL and EMD in patients with PCM. MIML was supervised by MJGV during the compilation of the results of this research work.

Point 4: TMTV calculation is difficult and the reproducibility is often suboptimal in many tumors but particularly in Multiple Myeloma: what was the agreement between the reviewers in TMTV calculation? It is important if you want to evaluate the prognostic value.

Response 4: As in Point 3, we did not considered it appropriate to calculate the agreement between the reviewers owing to the great difference in experience. We applied the recommended European Association of Nuclear Medicine threshold of SUVmax to segment the images [Reference 26] and explored other relative thresholds.

Point 5: The discussion is poor.

Response 5: Thank you. The discussion has been suitably modified.

Reviewer 2 Report

The authors presented paper "11C-Methionine PET/CT in Assessment of Multiple Myeloma Patients: Comparison to 18F-FDG PET/CT and Prognostic Value Evaluation"

1) The more fresh (2-3 years) references should be included in the introduction and discussion sections. 

2) Discussion section is limited. For the paper quality enhances I recommend enlarging discussion using quantitative values obtained in the work. The comparison with previous works will significantly improve the paper importance in the area.

3) Conclusion section is highly limited. I highly recommend enlarging it paying attention to the paper main results and novelty.

Minor comments

1) Abbreviations such as FDG, PET, CT, etc. should be disrupted for the wide range auditory of the Journal.

2) Keywords. I see some problems with keywords. I think it has to be much more.

3) Table 1. I recommend to reduce indentation in Table 1 to see the whole Table on one page.

Author Response

Thank you very much for your comments and observations.

As requested we have addressed all your comments. Changes in the revised version of the manuscript have been made using the "Track Changes" function of our word processing program. We hope that the revised version of the manuscript addresses all your concerns and those of the editor.

All authors have read and approved the revised version of the manuscript.

Looking forward to receiving your new comments and hopeful that the editor will find the manuscript suitable for publication.

Point 1: English language and style are fine/minor spell check required

Response 1: Spell-checking has been made according to British English.

Point 2: The introduction does not provide sufficient background and does not include all relevant references. Relevant more fresh (2-3 years) references to the research should be included in the introduction and discussion sections. 

Response 2: References 11- 18 have been removed and changed by more relevant references that expand the background and the discussion.

Point 3: Is the research design appropriate? Can be improved.

Response 3: The research design could be improved by including a larger series of patients with more progressions. But our interest was to show the medical community as soon as possible these interesting results and to stimulate other groups to design similar studies.

Point 4: Are the methods adequately described? Can be improved.

Response 4: Methods had been previously described in references 22 and 23, published by the authors of the present paper. However, if the reviewer finds it necessary to expand the description of this section, we would be grateful and will proceed.

Point 5: Are the results clearly presented? Must be improved.

Response 5: Indentation has been reduced in Table 1 in order to see the whole Table on one page.

In order to improve data analysis, we have decided to separate the sample population into 2 groups, NDMM and relapsed. We have performed the univariate analysis of relapsed patients, for a more comprehensive evaluation.

Point 6: Discussion section is limited.

Response 6: The discussion section has been enlarged with the quantitative values obtained in the work, and these values have been compared with previous publications.

Point 7: Are the conclusions supported by the results? Must be improved.

Response 7: The conclusion section has been enlarged paying attention to the paper main results and novelty.

Point 8: Abbreviations such as FDG, PET, CT, etc. should be disrupted for the wide range auditory of the Journal.

Response 8: MET and FDG PET/CT had been written instead of 11C-MET and 18F-FDG.

Point 9: I see some problems with keywords. I think it has to be much more.

Response 9: Relevant keywords had been added: multiple myeloma; positron emission tomography; PET/CT; 11C-methionine; 18F-FDG; prognosis; volume-based parameters; TMTV; TLG; TLMU.

Round 2

Reviewer 1 Report

The Authors responded adequately to the reviewer's comments

Reviewer 2 Report

Thank you for the revised paper.